# SARS-CoV-2 Lysate Stimulation Impairs the Release of Platelet-like Particles and Megakaryopoiesis in the MEG-01 Cell Line

**DOI:** 10.3390/ijms24054723

**Published:** 2023-03-01

**Authors:** Valentina Lopardo, Francesco Montella, Roberta Maria Esposito, Carla Zannella, Silvana Mirella Aliberti, Mario Capunzo, Gianluigi Franci, Annibale Alessandro Puca, Elena Ciaglia

**Affiliations:** 1Department of Medicine, Surgery and Dentistry “Scuola Medica Salernitana”, University of Salerno, Via Salvatore Allende, 84081 Baronissi, Italy; 2Department of Experimental Medicine, University of Campania “Luigi Vanvitelli”, 80138 Naples, Italy; 3Cardiovascular Research Unit, IRCCS MultiMedica, 20138 Milan, Italy

**Keywords:** SARS-CoV-2, platelets, MEG-01, megakaryopoiesis

## Abstract

SARS-CoV-2 infection causes a considerable inflammatory response coupled with impaired platelet reactivity, which can lead to platelet disorders recognized as negative prognostic factors in COVID-19 patients. The virus may cause thrombocytopenia or thrombocytosis during the different disease stages by destroying or activating platelets and influencing platelet production. While it is known that several viruses can impair megakaryopoiesis by generating an improper production and activation of platelets, the potential involvement of SARS-CoV-2 in affecting megakaryopoiesis is poorly understood. To this purpose, we explored, in vitro, the impact of SARS-CoV-2 stimulation in the MEG-01 cell line, a human megakaryoblastic leukemia cell line, considering its spontaneous capacity of releasing platelet-like particles (PLPs). We interrogated the effect of heat-inactivated SARS-CoV-2 lysate in the release of PLPs and activation from MEG-01, the signaling pathway influenced by SARS-CoV-2, and the functional effect on macrophagic skewing. The results highlight the potential influence of SARS-CoV-2 in the early stages of megakaryopoiesis by enhancing the production and activation of platelets, very likely due to the impairment of STATs signaling and AMPK activity. Overall, these findings provide new insight into the role of SARS-CoV-2 in affecting megakaryocyte–platelet compartment, possibly unlocking another avenue by which SARS-CoV-2 moves.

## 1. Introduction

Megakaryopoiesis is a complex stepwise process starting with hematopoietic stem cells that undergo multiple lineage commitments. Along the myeloid branch of hematopoiesis, immature megakaryocytes (MKs) differentiate into large polyploid megakaryocytes that eventually release platelets. Megakaryopoiesis and platelet production occurs in specialized bone marrow, osteoblastic and vascular niches rich in cytokines, chemokines, and growth factors, among which thrombopoietin (TPO) is the major regulator [1]. Approximately half of all platelet release a process termed *thrombopoiesis*, which occurs when megakaryocytes exit the bone marrow and enter the lung capillary beds, where they release platelets into circulation via proplatelet extensions [2,3]. Enrichment of lung MKs was described during pulmonary and cardiovascular diseases, further suggesting that they may be dynamic and responsive to inflammatory states [4,5]. Indeed, even though MK-derived platelets are primarily involved in vascular hemostasis, thrombosis, and wound healing, other related functions are emerging, including their role in immunity due to the plethora of inflammatory and bioactive molecules prevailing in platelet granules and released upon activation [6].

The activity and maturity of platelets can be discriminated according to their size as well as transcriptome and protein constitution. Larger platelets (diameter Ø 3–7 μm) are young and immature but more reactive and highly metabolically active, while smaller ones (Ø 0.5–3 μm) represent a terminally differentiated platelet sub-population [7]. Moreover, higher CD62p expression (e.g., platelet activation marker) and higher cytosolic content characterize the former [8].

Disruption of platelet homeostasis and megakaryocytic balance is commonly due to virus infection. Indeed, MKs and megakaryocytic cell lines are susceptible to several human pathogenic viruses. Dengue virus is able to interfere with megakaryopoiesis by affecting the activation of ERK1/2 MAP Kinase or STAT signaling, pathways that are known to be involved in megakaryocyte differentiation [1,9]. Furthermore, the Hantaan virus efficiently replicates in megakaryocytic cell lines, leading to the upregulation of MHC class I molecules [10]. In the same way, HIV infection of megakaryocytes induces MK impairment by altering the number of MKs with higher ploidy, leading to thrombocytopenia [11].

Different evidence showed that SARS-CoV-2 could directly interact with platelets or megakaryocytes through multiple and opposing mechanisms [12,13]. SARS-CoV-2 can be endocytosed by platelets or, more likely, platelets can be infected in the bone marrow, where, in turn, their MKs precursors are infected during maturation [14]. Furthermore, in SARS patients, the diffused alveolar damage and pulmonary endothelial cells may cause the activation and entrapment of platelets in the lungs, along with the thrombi formation at the sites of the injury, which may lead to the consumption of platelets [15,16]. In addition, as the lungs may also be the sites of platelet release from mature megakaryocytes, the damaged lungs may result in pulmonary fibrosis and pathological changes, affecting the lung megakaryocyte fragmentation and platelet production. In this context, patients with severe COVID-19 commonly experience coagulation disorders and platelet abnormalities associated with the severity of the disease [17].

The hinge between SARS-CoV-2/MK interaction and impairment in platelet production is poorly known, and its understanding could be beneficial to add another piece to the intricate COVID-19 physiopathology. In this light, we investigated the in vitro effect of SARS-CoV-2 in (a) altering the production of MK-derived platelet-like particles and (b) modifying the MK differentiation process. We used the MEG-01 cell line, a well-established model system for studying megakaryopoiesis and platelet-like particles [18]. Finally, we also hypothesized a putative mechanism by which SARS-CoV-2 could act, by exploring the STATs and AMPK signaling, well-known pathways influencing crucial steps of megakaryopoiesis and platelet production.

## 2. Results

### 2.1. SARS-CoV-2 Lysate Induces Immature MEG-01 to Release Platelet-like Particles

Several viruses (e.g., dengue virus, Zika virus, and human cytomegalovirus) can interfere with the process of megakaryopoiesis, generating an improper production and activation of platelets that mainly result in severe platelet disorders [19,20,21]. The less-known involvement of SARS-CoV-2 in megakaryopoiesis prompted us to evaluate the effect of heat-inactivated lysate from SARS-CoV-2-infected cells (5 μg/mL) on MEG-01, a megakaryoblastic cell line widely used for the spontaneous release of platelet-sized particles into the culture medium and for sporadic cytoplasmic processes that make it comparable to megakaryocyte proplatelets [22,23]. Of course, in all experiments, the results were compared to those obtained in cells exposed to control lysate (5 μg/mL) produced from non-infected cells in the same conditions, herein indicated as CTR.

We evaluated the effect in both undifferentiated and differentiated MEG-01 cells, the latter obtained upon the appropriate treatment with phorbol 12-myristate 13-acetate (PMA 5 nM) and thrompoietin (TPO 100 ng/mL) for 72 h. The application of an automated cell counter allowed us to discriminate three cell subpopulations: differentiated platelet-like particles (Ø 1–3 μm), larger and immature platelet-like particles (Ø 4–7 μm), and larger cells representing proper megakaryocyte-like cells (Ø 16–35 μm). After 3 h and 24 h of stimulation with SARS-CoV-2 lysate (5 μg/mL), undifferentiated MEG-01 cells were led to release platelet-like particles while the production of megakaryocytes was prevented (Figure 1A,B). Of note, when differentiated MEG-01 cells were exposed to SARS-CoV-2 lysate (5 μg/mL), platelet-like particles decreased, but the virus lysate did not affect the megakaryocyte compartment (Figure 1C,D). To corroborate the obtained results, we moved to characterize the platelet-like and MK subpopulations through flow cytometry analysis. As expected, by gating platelets based on forward scatter (FSC) and side scatter (SSC) properties and the positivity for the platelet double marker CD41/CD61 to exclude MK cells and debris, SARS-CoV-2 lysate (5 μg/mL) increased the expression of CD41+/CD61+ cells in undifferentiated MEG-01 (Figure 1E). As expected, we detected a lower percentage of megakaryocytes such as ESAM+CD61- cells in parental MEG-01 exposed to SARS-CoV-2 lysate (5 μg/mL) for 24 h, which confirmed the above-detailed cell count (Figure 1G,H).

Taken together, these first results suggest that SARS-CoV-2 lysate could boost the release of platelets mainly from immature megakaryocytes and prevent the recovery of the megakaryocytic cell pool.

### 2.2. SARS-CoV-2 Lysate Interferes with Platelet-like Particle Activation but Not with Cell Viability

Patients with COVID-19 experience thrombotic events mainly due to platelet hyperactivation that is achieved by the release of granule-stored mediators [24]. To test if SARS-CoV-2 lysate was capable of hyperactivating the platelet-sized particles besides its ability to massively induce the release of platelet-like particles from undifferentiated MEG-01, we evaluated the expression of the platelet activation markers CD107a (or LAMP-1) and CD62p (or P-selectin) in undifferentiated MEG-01 cells in resting condition and after 3 h of virus lysate stimulation. In our in vitro experimental setting, we found that SARS-CoV-2 lysate (5 μg/mL) was able to elicit a statistically significant increase in the expression of both surface markers CD107a and CD62p uniquely in undifferentiated MEG-01 cells (Figure 2A), with no effect on the differentiated MEG-01 (Figure 2B).

To ascertain whether virus lysate interferes with the viability of parental MEG-01, we performed Annexin V/propidium iodide staining through FACS analysis. We did not collect any apoptosis evidence either at the early or late apoptotic stages, as shown in the bar graph reported in Figure 2C. In support of this, we also verified whether SARS-CoV-2 lysate was able to affect the mitochondrial activity of parental MEG-01 through the flow cytometric analysis of the MitoBright fluorescence signals. Considering that the percentage of MitoBright-positive SARS-CoV-2-exposed MEG-01 cells was comparable to that of resting condition cells, we deduced that SARS-CoV-2 lysate did not induce mitochondrial depolarization events, an early sign of MK apoptosis (Figure 2D).

### 2.3. SARS-CoV-2 Lysate Impairs Megakaryopoiesis through STATs Signaling and AMPK Activity

Megakaryopoiesis is supported by the activation of JAK-STAT intracellular signaling [25]. Thus, we wondered whether the JAK-STATs modulation accompanied the megakaryopoiesis impairment carried out by SARS-CoV-2 lysate on MEG-01. We found that SARS-CoV-2 lysate was capable of preventing the phosphorylation of Signal Transducer and Activator Of Transcription 1 (STAT-1) in parental MEG-01 (Figure 3A). Moreover, since multiple viruses affect megakaryopoiesis by altering the host cell metabolism, we also examined certain signal molecules of the PI3K/Akt and AMPK metabolic pathways, as shown in Figure 3B [26,27]. SARS-CoV-2 lysate led to increased phosphorylation of AKT coupled with the decreased expression of phosphorylated AMPKα in naive MEG-01 with respect to MEG-01 in a resting condition.

To validate our experimental setting, we also investigated the megakaryopoiesis-related pathway and the metabolic signaling in MEG-01 infected with live SARS-CoV-2. As shown in Figure 3B SARS-CoV-2 replication in vivo induced the phosphorylation of STAT-1 as found in cells exposed to heat-inactivated virus lysate. Moreover, SARS-CoV-2 replication dysregulated AMPK intracellular signaling by significantly decreasing the expression of p-AMPKα, as found in the heat-inactivated SARS-CoV-2 cell model. By contrast, SARS-CoV-2 replication did not increase the phosphorylated amount of AKT, as highlighted for the inactivated SARS-CoV-2, probably due to the different virus kinetics of replication and infectivity [28].

### 2.4. The Conditioned Medium from SARS-CoV-2-Treated MEG-01 Predisposes Macrophages to Acquire an Anti-Inflammatory Phenotype

Viruses evolved several strategies to escape the antiviral responses to achieve a supportive microenvironment for viral replication in host cells [29]. According to this, we found that SARS-CoV-2 lysate stimulated IL-1β secretion and reduced the antiviral IL-8 release from parental MEG-01 cells (Figure 4A). To determine whether the conditioned medium (CM) from SARS-CoV-2-treated MEG-01 was effective in inducing immune cells to acquire an antiviral phenotype, we conditioned healthy donors’ monocytes with supernatant harvested from control and SARS-CoV-2-treated MEG-01. As shown in Figure 4B, after 7 days of treatment, we harvested a heterogenous cell population of both M1 (CD163+CD86+) and M2 (CD163+CD206+) macrophages. Specifically, we found that releasates from SARS-CoV-2-treated MEG-01 led monocytes to skew into M2-like macrophages and to properly downregulate M1 markers. In view of this, we observed a statistically significant reduction in the M1/M2 ratio after exposing monocytes to the CM of SARS-CoV-2-treated MEG-01 with respect to resting conditions.

## 3. Discussion

COVID-19 is an emerging infectious disease, and its hematological changes are common [30,31,32]. Elucidation of the mechanisms of how SARS-CoV-2 infection causes these alterations is important for better management of the disease. In the current study, we provided evidence that SARS-CoV-2 could impair the process of megakaryopoiesis and affect the formation of platelets in MEG-01 cells, a megakaryoblastic cell line that proved to be a useful model for studying megakaryopoiesis and platelet activity. Indeed, MEG-01 cells (a) have morphological and phenotypical features resembling those of megakaryocytes, (b) spontaneously release platelet-like particles, and (c) can be differentiated by PMA and TPO into mature megakaryocytes that we here defined differentiated MEG-01. These properties allowed us to reproduce a quite accurate in vitro experimental model where both naive and differentiated MEG-01 cells were exposed to the heat-inactivated SARS-CoV-2 lysate. Viral lysates, derived from infected cells that have undergone inactivation, are becoming a valuable tool in the study of viral proteins and their ability to elicit immune responses [33,34].

Firstly, we found that the virus lysate triggers a significant release of both immature and small platelet-like particles in parental MEG-01 cells, while differentiated MEG-01 cells are refractory to its effects. Concerning the mechanism, as reported by others, increased platelet counts might be driven by inappropriate cytokine production during infection and chronic inflammation, as well as in COVID-19 [35,36,37]. Common mediators of reactive thrombocytosis include IL-1β, IL-6, GM-CSF, TNF-α, and IFN-γ [36]. Consistent with this, we reported that the conditioned medium from SARS-CoV-2-treated MEG-01 is enriched for IL-1β (Figure 4), which might predispose to and sustain enhanced thrombopoiesis.

Conversely, SARS-CoV-2 lysate can prevent the recovery of the ESAM+ megakaryocytic cells, as achieved after exposing undifferentiated MEG-01 to the virus lysate for 24 h (Figure 1A,B). Since the differentiated MEG-01 cells were refractory to the virus lysate effects, it can be speculated that PMA-TPO pretreatment of cells, leading to the differentiation, resulted in an altered protein expression affecting key molecules with a putative role in cell surface receptor for lysate or a downstream step. However, this was not investigated further. At a functional level, platelet-like particles released by SARS-CoV-2-exposed parental MEG-01 are hyperactivated, considering their increased expression of lysosomal-associated membrane protein (LAMP-1) and P-selectin on the cell surface. In this way, SARS-CoV-2 lysate may evoke early exocytosis of lysosomal contents, known to participate in vessel wall remodeling and a consequent thrombo-inflammatory state [38]. Indeed, Barret and colleagues reported that the exposure of endothelial cells to platelet releasates from COVID-19 patients could induce an inflammatory hypercoagulable endotheliopathy of prognostic relevance [39].

Collectively, our data could hint at another pathophysiological mechanism underlying the thrombotic disorders in COVID-19 patients. Indeed, SARS-CoV-2 could evoke a massive response in megakaryocytes through the release of hyperactivated young platelets that could significantly contribute to the pro-thrombotic platelet activities in an early stage of the disease since immature hyper-reactive platelets are associated with arterial thrombotic events [39,40]. At the same time, in our experimental setting, SARS-CoV-2 was capable of reducing the platelet release from mature megakaryocytes, in line with severe thrombocytopenia experienced by high-grade COVID-19 patients [41]. Even though the receptors for the virus lysate are currently unknown, these findings indicate that viral proteins play a unique role as regulators of the megakaryocyte lineage by simultaneously influencing the generation and activation of platelets and the blockade of the megakaryocyte maturation. Indeed, lysate stimulation of MEG-01 causes striking proplatelets burst, detected after only 3 h, which is not properly sustained by enrichment of megakaryocyte numbers.

From a mechanistic point of view, as we conducted isolated experiments in an in vitro cell culture system, we might hypothesize that an aberrant platelet production in bone marrow or in the lung may be a more important mechanism than platelet clearance or their enhanced consumption during COVID-19 to explain thrombopoiesis impairment by the virus lysate. Indeed, despite the MK turnover in the bone marrow niche or in the lung needing a more adequate model, our findings show plausible evidence that SARS-CoV-2 may directly hinder the early stages of the megakaryopoiesis process and is not necessarily a result of diffused lung damage and fibrosis, advocated as a potential mechanism of thrombocytopenia [42]. At a molecular level, the inhibition of STAT-1, a well-established regulator of megakaryopoiesis [43], in immature megakaryocytes sustains our hypothesis according to which SARS-CoV-2 targets undifferentiated megakaryocytes to induce an imbalance in platelet production.

Interestingly, we also reported a SARS-CoV-2 lysate-mediated impairment of the AMPK-AKT metabolic axis. Due to its key role in cell homeostasis, activated AMPK is an essential signaling molecule that many viruses utilize to replicate [44]. On the other hand, activated AMPK potentiates the antiviral host defenses, and many viruses have developed mechanisms to inhibit AMPK. Viruses such as HCV and HIV inhibit AMPK to enhance their replication [45,46]. For other viruses, such as DENV, CVB3, and influenza virus, there are contradicting reports [27,47,48]. In our experimental model, SARS-CoV-2 lysate stimulation impairs the crosstalk between PI3k/Akt and AMPK signaling on naive MEG-01. As reported by others, some regulatory sites in AMPK subunits can be phosphorylated by kinases, such as Akt, implicated in the negative regulation of AMPK. Here, we show that SARS-CoV-2 inhibits the activity of AMPK and lessens the MEG-01 cell metabolism by increasing the phosphorylation of Akt, which may influence AMPK through a counterregulatory mechanism. Consistent with our results, a similar regulatory mechanism has been reported for the dengue virus in megakaryocytes or Zika virus in endothelial cells [26,27].

As we used a heat-inactivated virus lysate, able to replicate in MEG-01, we might hypothesize that it affects the megakaryocyte AMPK state, not for viral replication but probably to impair cell homeostasis and immune regulatory functions of MK cells. Indeed, AMPK is a key energy regulator not only for cell growth, proliferation, and stress responses but also for host immune regulatory function [44]. In this context, we found that SARS-CoV-2 lysate burst leads immature megakaryocytes to develop a favorable microenvironment consisting of increased pro-inflammatory factors and decreased antiviral mediators, respectively, IL-1β and IL-8. Since platelets store inflammatory mediators in granules and we found that SARS-CoV-2 stimulates the release of platelet-like particles from naive MEG-01, we suggest that SARS-CoV-2 shapes the surrounding environment by triggering the release of platelet granule-stored mediators committed to the immune response. Macrophages represent a first line of defense against infection. In support of the above-mentioned virus features, we first described that conditioned medium CM from SARS-CoV-2-triggered MEG-01 could imbalance the M1/M2 ratio on behalf of the anti-inflammatory and myelosuppressive M2-phenotype. This might be expected if we consider that viruses and several variants have a combined evolution of antibody escape with enhanced antagonism of human innate immunity to improve transmission and possibly reduce immune protection from severe disease [49,50].

Even though future studies will aim to understand the interaction of SARS-CoV-2 lysate with MK and platelet compartment and how the crosstalk between platelet and innate immunity occurs, our findings shed light on MKs as a novel SARS-CoV-2 infection target. Furthermore, the identification and better underpinning of altered signaling pathways such as STAT-1 and AKT/AMPK could have several therapeutical implications, not only in COVID-19 but also in other thrombotic states associated with a wide spectrum of infectious diseases.

## 4. Materials and Methods

### 4.1. SARS-CoV-2 Lysate Production

Virus lysate was collected from MEG-01-infected cells (MOI 1). Cells were cultured in T75 flasks (Thermo Fisher Scientific, Waltham, MA, USA) until they reached 80% confluence. Twenty-four hours post-infection with lytic viruses destroying cells they invade, cell supernatant was harvested and centrifuged at 1200 rpm at 4 °C for 10 min in the presence of 0.5% Triton X 100 plus 0.6 M KCl [51]. Finally, the virus lysate was inactivated at 95 °C for 5 min and stored at −80 °C [52]. All experimental work involving SARS-CoV-2 was performed in a biosafety level 3 (BSL3) containment laboratory. SARS-CoV-2 clinical isolate was kindly donated by Lazzaro Spallanzani Hospital, Rome, Italy. 

### 4.2. Cell Line and Culture Condition

MEG-01 (ATCC^®^ CRL-2021) cells were grown in a humidified incubator at 37 °C, and 5% CO_2_ in RPMI-1640 (Gibco^®^, Thermo Fisher Scientific, Waltham, MA, USA) supplemented with 10% (*v*/*v*) fetal serum bovine (FBS, Gibco^®^, Thermo Fisher Scientific, Waltham, MA, USA), 1% (*v*/*v*) penicillin-streptomycin (Aurogene, Rome, Italy), 1% (*v*/*v*) MEM non-essential amino acids (MEM NEAA, Gibco^®^, Thermo Fisher Scientific, Waltham, MA, USA), and 1% (*v*/*v*) sodium pyruvate (Aurogene, Rome, Italy). Differentiated MEG-01 cells were obtained upon 3 days of pretreatment with phorbol 12-myristate 13-acetate 5nM and thrompoietin 100 ng/mL. Both naive and differentiated MEG-01 cells were exposed to heat-inactivated SARS-CoV-2 lysate (5 μg/mL) for up to 24 h. All the results were compared to the effects of cells exposed to control lysate (5 μg/mL), here indicated as CTR, obtained from non-infected MEG-01 in the same conditions. At different time points, supernatants and/or cells were collected for subsequent assays.

### 4.3. Platelets Count

Undifferentiated and differentiated MEG-01 were plated into a 12-well plate (150.000 cells/well) and treated as above-mentioned. After 3 h and 24 h stimulation, cells were harvested and counted. By using an automatic cell counter (LUNA Automated Cell counter, logos), three cell fractions were considered: small platelets (Ø 1–3 μm, regarded as terminally differentiated MEG-01, and so platelet-like particles), large platelets (Ø 4–7 μm, regarded as young, immature platelet-like particles), and megakaryocytes (Ø 16–35 μm, mean 21.8 μm; referred to as mature MK).

### 4.4. Cytokine Detection

The cytokine levels in SARS-CoV-2 exposed MEG-01-conditioned media were determined using a bead-based multiplex ELISA (LEGENDplexTM, Biolegend, San Diego, CA, USA). Supernatants were incubated for 2 h with the beads and for 1 h with the detection antibodies, followed by 30 min incubation with SA-PE. After washing, beads were resuspended in washing buffer and acquired using a FACS VERSE flow cytometer (BD Biosciences, Franklin Lakes, NJ, USA). Data were analyzed with the LEGENDplex Data Analysis Software.

### 4.5. Flow Cytometry Analysis

#### 4.5.1. Antibodies

MEG-01 cells (treated as above-mentioned) were stained with mAb against human CD62p (Miltenyi Biotec, Bergisch Gladbach, Germany), CD107a (Miltenyi Biotec, Bergisch Gladbach, Germany), CD61 PE (Miltenyi Biotec, Bergisch Gladbach, Germany), and CD41/CD61 (Miltenyi Biotec, Bergisch Gladbach, Germany). After 30 min incubation at 4 °C in the dark, cells were washed with staining buffer (PBS 2% fetal serum bovine, 0.01% sodium azide), centrifuged at 1200 rpm for 5 min, and resuspended in staining buffer for the FACS analysis. For each test, cells were analyzed using a FACS VERSE flow cytometer (BD Biosciences, Franklin Lakes, NJ, USA).

#### 4.5.2. Apoptosis Assay

An FITC Annexin V Apoptosis Detection Kit with PI (BioLegend) was employed to detect cell necrosis and apoptosis in MEG-01 exposed to inactivated SARS-CoV-2 lysate (10 μg/mL) for 18 h. After washing, cells were resuspended in Annexin V Binding solution and incubated with Annexin V-FITC and PI solution for 15 min at RT in the dark. Stained cells were resuspended in Annexin V Binding solution and acquired using a FACS VERSE flow cytometer (BD Biosciences, Franklin Lakes, NJ, USA).

#### 4.5.3. Mitochondrial Activity

MitoBright LT Red (Dojindo Technology, Rockville, MD, USA) was applied in order to evaluate the mitochondrial activity in MEG-01 exposed to inactivated SARS-CoV-2 lysate (5 μg/mL) for 18 h according to the manufacturer’s protocol (Dojindo Technology). At the end of the treatment, cells were trypsinized, resuspended in Hank’s Balanced Salt Solution (HBSS, Sigma Life Science, St. Louis, MO, USA), and incubated with 100 nM MitoBright Red solution for 30 min at 37 °C in the dark. After washing, stained cells were resuspended in HBSS, and the fluorescence of the cells was then analyzed over time using a FACS VERSE flow cytometer (BD Biosciences). The mean fluorescence intensity (MFI) for each sample was normalized to that of a control sample to calculate the relative fluorescence intensities for the samples [21].

### 4.6. Western Blotting

MEG-01 cells were washed with PBS (Gibco^®^, Thermo Fisher Scientific), harvested, and lysed in ice-cold RIPA lysis buffer (50 mM Tris-HCl, 150 mM NaCl, 0.5% Triton X-100, 0.5% deoxycholic acid, 10 mg/mL leupeptin, 2 mM phenylmethylsulfonyl fluoride, and 10 mg/mL aprotinin) as also detailed by Ciaglia et al. [53]. After centrifugation (13,000 rpm for 20 min at 4 °C), in order to remove the cell debris, proteins were quantified. About 30 μg of proteins were separated on 10% SDS-PAGE at 90 V for 1 h and at 120 V for 1 h and then transferred to a nitrocellulose membrane. After blocking with 5% nonfat dried milk powder (PanReac AppliChem) in Tris-buffered saline containing 0.1% Tween-20 (TBST) for 1 h at room temperature, the membranes were incubated overnight with the following primary antibodies: anti-p-STAT1 (Y701, Cell Signaling Technology #7649, rabbit mAb 1:1000), anti-p-AMPKα (T172, Cell Signaling Technology #2535, rabbit mAb 1:1000), anti-AMPKα (Cell Signaling Technology #2603, rabbit mAb 1:1000), anti-p-AKT (S743, Thermo Fisher Scientific #700392, rabbit mAb 1:1000), and anti-AKT (Cell Signaling Technology #4691, rabbit mAb 1:1000). Immunodetection of specific proteins was carried out with horseradish peroxidase-conjugated donkey anti-mouse or anti-rabbit IgG (Bio-Rad), using the enhanced chemiluminescence (ECL) system (Thermo Fisher Scientific) according to the manufacturer’s instructions and then exposed to X-ray films (Thermo Fisher Scientific). Western-blot data were analyzed using Photoshop software to determine the optical density (OD) of the bands. The OD readings of phosphorylated proteins were expressed as a ratio relative to the total protein and/or β-actin.

### 4.7. Statistical Analysis

In all the experiments shown, statistical analysis was performed by using the GraphPad Prism 6.0 software for Windows (GraphPad software). For each type of assay or phenotypic analysis, data obtained from multiple experiments are calculated as mean ± SD and analyzed for statistical significance using appropriate tests. In an analysis of variance (ANOVA) for multiple comparisons, *p*-values < 0.05 were considered significant; * *p* < 0.05, ** *p* < 0.01, and *** *p* < 0.001.

## Figures and Tables

**Figure 1 ijms-24-04723-f001:**
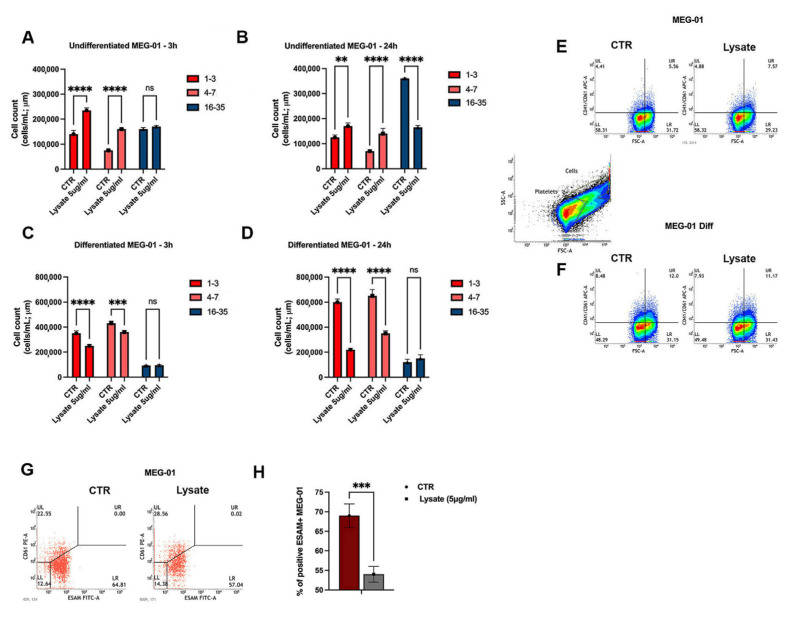
SARS-CoV-2 lysate effect on platelet-like particles and megakaryocytes in undifferentiated and differentiated MEG-01. (**A**–**D**) The histograms show the effect of inactivated SARS-CoV-2 lysate (5 μg/mL) compared to those of control lysate (CTR) on undifferentiated (**A**,**B**) and differentiated MEG-01 (obtained upon 72 h treatment with thrombopoietin TPO 100 ng/mL and phorbol 12-myristate 13-acetate PMA 5 nM); (**C**,**D**) exposed to virus lysate for 3 h and 24 h, considering three subpopulations, discriminated by using an automized cell counter: red bars represent terminally differentiated platelet-like particles (Ø 1–3 μm), pink bars shows larger and immature platelet-like particles (Ø 4–7 μm), and blue bars represent proper megakaryocytes (Ø 16–35 μm). While inactivated SARS-CoV-2 lysate (5 μg/mL) induces the release of platelet-like particles in undifferentiated MEG-01 at both early and late stages of stimulation with the megakaryocyte production blockade, it prevents the platelet-like particles in differentiated MEG-01. All the results were compared to the effects of cells exposed to control lysate CTR (5 μg/mL) obtained from non-infected MEG-01 in the same conditions. The results are representative of three independent experiments expressed as mean ± SD. Pairwise comparisons are reported (ordinary one-way ANOVA; ** *p* < 0.01, *** *p* < 0.001, **** *p* < 0.0001, *ns* = no significant *p*-values were found). (**E**,**F**) Panel shows representative density plots highlighting the expression of CD41/CD61-positive cells through flow cytometry upon an appropriate gating strategy in order to uniquely acquire platelet-like particles and characterize them in undifferentiated (**E**) and differentiated MEG-01 (**F**) exposed to inactivated SARS-CoV-2 lysate (5 μg/mL) for 24 h. (**G**) Representative dot plot of flow cytometry analysis of ESAM expression in undifferentiated MEG-01 cells exposed to inactivated SARS-CoV-2 lysate (5 μg/mL) for 24 h upon an appropriate gating strategy based on the CD61 expression. (**H**) The bar graph shows the percentage of positive ESAM+ megakaryocytes in undifferentiated MEG-01 cells exposed to inactivated SARS-CoV-2 lysate (5 μg/mL) for 24 h. The result is representative of three independent experiments expressed as mean ± SD. A pairwise comparison is reported (ordinary one-way ANOVA; *** *p* < 0.001).

**Figure 2 ijms-24-04723-f002:**
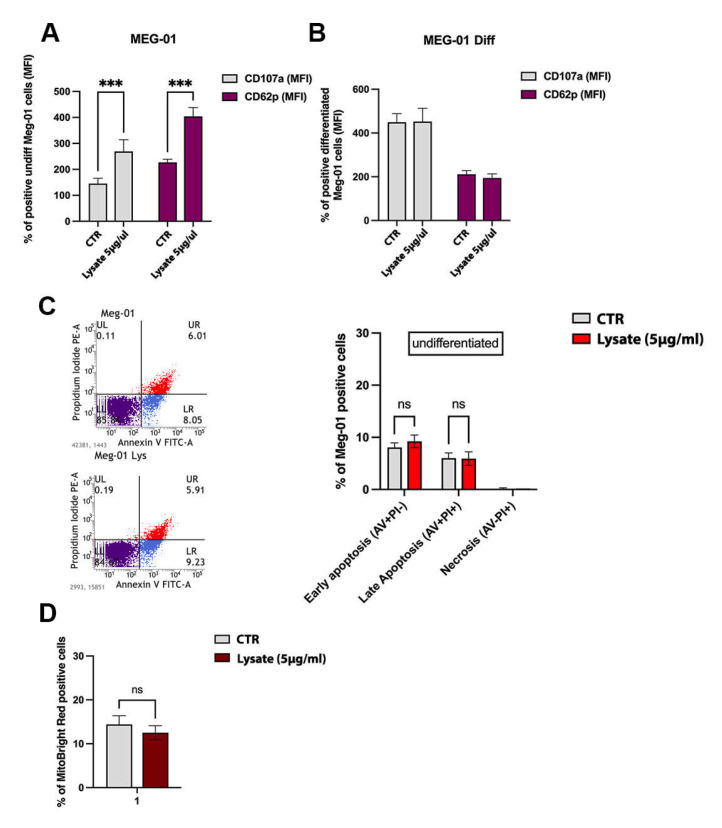
Functional characterization of platelet-like particles released by MEG-01 cells. (**A**,**B**) SARS-CoV-2 lysate’s effect on platelet-like particle activation. The bar graphs show the percentage of CD107a+ (gray bars) and CD62p+ (purple bars)-positive undifferentiated and differentiated MEG-01 cells exposed to inactivated SARS-CoV-2 lysate (5 μg/mL) or control lysate for 3 h. The cytofluorimetric mean fluorescence intensity (MFI) is indicated. The result is representative of three independent experiments expressed as mean ± SD. Pairwise comparison is reported (ordinary one-way ANOVA; *** *p* < 0.001; in (**B**), no significant *p*-values were found, reported as ns). (**C**) SARS-CoV-2 lysate induction of apoptosis measured by Annexin V and propidium iodide (PI) double-staining through flow cytometry in the undifferentiated MEG-01 cell line after 24 h of treatment with inactivated SARS-CoV-2 lysate (5 μg/mL). Histograms on the right indicate the total percentage of early (Annexin V-positive cells/PI-negative cells) and late apoptotic events (Annexin V/PI-double-positive cells) as well as necrotic cells (Annexin V-negative cells/PI-positive cells). The results are representative of three independent experiments expressed as mean ± SD. Statistical analysis by two-way ANOVA with Tukey’s test for multiple comparisons was conducted (no significant *p*-values were found). (**D**) SARS-CoV-2′s impact on mitochondrial activity was measured by MitoBright Red staining by flow cytometry in the undifferentiated MEG-01 cell line after 24 h of treatment with inactivated SARS-CoV-2 lysate (5 μg/mL). The bar graph shows the percentage of MitoBright Red-positive parental MEG-01 cells as a result of three independent experiments expressed as mean ± SD. All the results were compared to the effects of cells exposed to control lysate CTR (5 μg/mL) obtained from non-infected MEG-01 in the same conditions. A pairwise comparison is reported (ordinary one-way ANOVA; no significant *p*-value was found).

**Figure 3 ijms-24-04723-f003:**
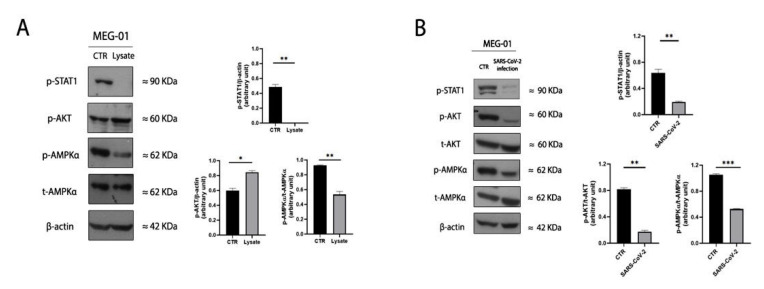
SARS-CoV-2 stimulation alters the megakaryopoiesis pathway and AMPK activity in undifferentiated MEG-01. (**A**) Panel shows immunoblot bands and relative quantification of control and inactivated SARS-CoV-2-exposed whole-cell lysates of undifferentiated MEG-01 cells immunoblotted for p-STAT1, p-AKT, p-AMPKα, total AMPKα, and β-actin. On the right side, bar graphs represent the expression levels of p-STAT1, p-AKT, and p-AMPKα quantified by densitometry and normalized to the respective loading control β-actin. Pairwise comparisons are reported (ordinary one-way ANOVA; * *p* < 0.05, ** *p* < 0.01, *** *p* < 0.001). The results are representative of three independent experiments expressed as mean ± SD. (**B**) Panel shows control and SARS-CoV-2-infected whole-cell lysates of undifferentiated MEG-01 cells immunoblotted for p-STAT1, p-AKT, total AKT, p-AMPKα, total AMPKα, and β-actin. On the right side, bar graphs represent the expression levels of p-STAT1, p-AKT, and p-AMPKα, quantified by densitometry. The band densities of the proteins were normalized to the respective loading control β-actin or to the respective total protein. Pairwise comparisons are reported (ordinary one-way ANOVA; ** *p* < 0.01, *** *p* < 0.001, no significant *p*-values are not shown). The results are representative of three independent experiments expressed as mean ± SD.

**Figure 4 ijms-24-04723-f004:**
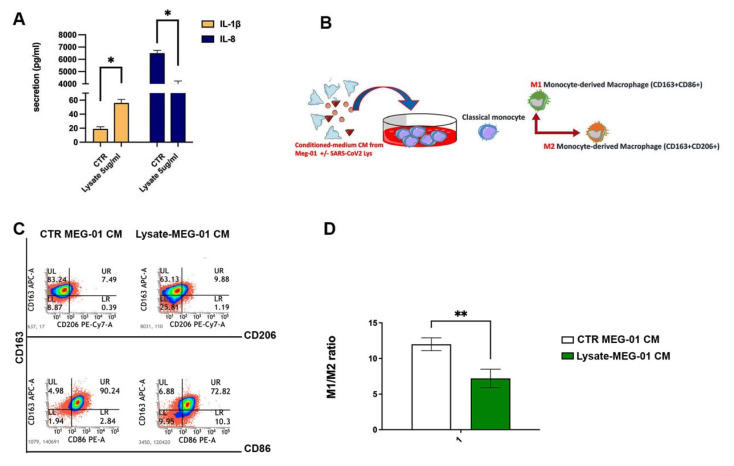
SARS-CoV-2 lysate affects platelet-like particles/MEG-01 secretome and M1/M2 ratio. (**A**) Histogram shows IL-1β and IL-8 secretion levels (pg/mL) in conditioned media of undifferentiated MEG-01 exposed to inactivated SARS-CoV-2 (5 μg/mL) or control lysate for 24 h and detected by multiplex ELISA. Pairwise comparisons are reported (ordinary one-way ANOVA; * *p* < 0.05). The results are representative of three independent experiments expressed as mean ± SD. (**B**) Graphical diagram summarizing the steps of the applied protocol. Conditioned media from undifferentiated MEG-01 exposed or not exposed to inactivated SARS-CoV-2 (5 μg/mL) lysate were added to classical monocytes obtained from donors’ whole blood to induce M1 and M2-like macrophages. After 7 days, macrophages were harvested and characterized through flow cytometry. M1-monocyte-derived macrophages are CD163/CD86 double-positive cells, while M2-monocyte-derived macrophages are CD163/CD206 double-positive cells. (**C**,**D**) Representative FACS dot plots displaying the abundance of different macrophage subsets based on the expression of CD163, CD86, and CD206 markers expressed at the cell surface of recovered macrophage upon 7 days of conditioning with supernatant from MEG-01 exposed or not exposed to inactivated SARS-CoV-2 (5 μg/mL) lysate. Relative mean fluorescence intensity (MFI) ± SD of three independent experiments is shown as the M1/M2 (CD163+CD86+/CD163+CD206+) ratio (**D**). A pairwise comparison is reported (one-way ANOVA, ** *p* < 0.01).

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
