# Peer review of "SARS-CoV-2 Lysate Stimulation Impairs the Release of Platelet-like Particles and Megakaryopoiesis in the MEG-01 Cell Line"

_ijms, 2023, doi:10.3390/ijms24054723_

Round 1

Reviewer 1 Report

In this study Leopardi et al. showed that in undifferentiated MEG-01, but not in differentiated-ones, SARS-CoV-2 lysate induces significant release of platelet-like particles (after 3 and 24 hrs of incubation) with inhibitory effects on megakaryocytes generation (only after 24 hrs of incubation). This phenomenon is mainly accompanied by platelet activation with no interference on cell viability (as demonstrated by the increase of platelet activation markers and no effect on mitochondrial activity), involvement of JAK-STAT, PI3K/Akt, AMPK signalings, and increased release of the inflammatory cytokine IL-1beta and decreased release of the anti-viral IL-8.

This paper is interesting, clear, well written and conceived. However, the Reviewer has some concerns.

1)The nature and composition of SARS-CoV-2 lysate used to stimulate MEG-01 cell line are not clear. Please, expand its description.

2)A heat-inactivated virus lysate rules out an effect dependent of virus replication but which is the role of spike protein in influencing megakaryopoiesis in this study?

3)The Authors should explain the crosstalk between PI3K/Akt and AMPK pathway. Which is the effect of  inhibition of PI3K/Akt pathway (for instance with wortmannin) on AMPK and platelet release from megakaryocytes?

Author Response

We thank the reviewer for his/her positive remarks and we are pleased to know that he/she considered our results interesting, clear and well conceived. We also thank the reviewer for his/her meaningful suggestions on how to improve our scientific report. 

In particular we re-examined point by point, all the issues and corrections raised by the reviewer and appropriately modified the manuscript.

1) The nature and composition of SARS-CoV-2 lysate used to stimulate MEG-01 cell line are not clear. Please, expand its description.

We added a better description of virus lysate preparation in Materials and Methods section (line 388-395) and discussed the adjunct value deriving from the use of virus lysate. Indeed viral lysates, derived from infected cells that have undergone inactivation, are a valuable tool in the study of viral proteins and their ability to elicit immune responses (Wang Q. et al, Functional properties of the spike glycoprotein of the emerging SARS-CoV-2 variant B.1.1.529. Cell Rep. 2022 Jun 14;39(11):110924; Lavie M. et al, SARS-CoV-2 Spike Furin Cleavage Site and S2' Basic Residues Modulate the Entry Process in a Host Cell-Dependent Manner. J Virol. 2022 Jul 13;96(13):e0047422).

Our specific SARS-CoV-2 lysate, obtained from MEG-01 cells, retains the full complement of proteins present in non-inactivating viral suspensions and has been utilized to investigate: 1) the spike glycoprotein of emerging variants (Jeyaraman M. et al, Platelet lysate for COVID-19 pneumonia-a newer adjunctive therapeutic avenue. Stem Cell Investig. 2021 Jun 8;8:11); 2) the modulation of viral entry through spike furin cleavage sites and S2' basic residues (Langnau C. et al, Platelet Activation and Plasma Levels of Furin Are Associated With Prognosis of Patients With Coronary Artery Disease and COVID-19. Arterioscler Thromb Vasc Biol. 2021 Jun;41(6):2080-2096); 3) the roles of platelets in cell death, antiviral response and viral replication in COVID-19 (Trugilho MRO. et al, Platelet proteome reveals features of cell death, antiviral response and viral replication in covid-19. Cell Death Discov. 2022 Jul 16;8(1):324); 4) platelet-mediated amplification of endotheliopathy in COVID-19 (Barrett TJ. Et al, Platelets amplify endotheliopathy in COVID-19. Sci Adv. 2021 Sep 10;7(37):eabh2434).

2)A heat-inactivated virus lysate rules out an effect dependent of virus replication but which is the role of spike protein in influencing megakaryopoiesis in this study?

We thank the reviewer for his/her positive valuable consideration. Indeed as clearly reported, a heat-inactivated virus lysate certainly rules out an effect dependent of virus replication. Concerning the effect of spike protein in our cellular model in vitro, we focused our attention on the role of virus lysate and not spike protein alone for two main reasons:

i) several immunofluorescence analysis and western blot assays showed a lack of ACE2 expression, the conventional receptor for Spyke protein, in both human platelets and megakaryocytes as documented elsewhere (Shen, S., Zhang, J., Fang, Y. et al. SARS-CoV-2 interacts with platelets and megakaryocytes via ACE2-independent mechanism. J Hematol Oncol 14, 72 (2021). https://doi.org/10.1186/s13045-021-01082-6). Indeed many receptors other than ACE2 have been described to interact with a plethora of viral proteins and affect platelets or megakaryocytes. In MEG-01 cells, CD147, GRP78, KREMEN1, cathepsin L, NRP1, and ASGR1 were detected, while in platelets, CD147, GRP78, KREMEN1, and ASGR1 were detected;

ii) then we decided to use virus lysate mainly because it can mimic the immunomodulatory action of cellular debris and virus lysate derived from cellular disfunction and death causing multi-organ failure in COVID-19.

For all these reasons we believe that our specific SARS-CoV-2 lysate, which retains the full complement of proteins present in non-inactivating viral suspensions may have adjunct value to target MK compartment compared to spike protein alone.

3) The Authors should explain the crosstalk between PI3K/Akt and AMPK pathway. Which is the effect of inhibition of PI3K/Akt pathway (for instance with wortmannin) on AMPK and platelet release from megakaryocytes?

Eventhough the pharmacological inhibition of PI3k/Akt pathway is a valuable suggestion, we didn’t check it in our previous data because the role of AKT is a well-established mechanism leading to platelets ‘release. Unfortunately, the scarcity of available material (virus lysate) and the time-consuming production of our SARS-CoV-2 virus lysate led us to focus on selected experiments with evidence of experimental novelty effects. Indeed, Lahon et al (doi: 10.3389/fcimb.2021.715208) already showed the inhibition of PI3k/Akt pathway by Akt inhibitor IV (5uM) and PI3K inhibitor LY294002 (10 uM) abolished the expression of transcription factors involved in megakaryopoiesis and reduced the expression of CD61 in MEG-01 cells exposed to Dengue virus. Moreover, as reported by Guerriero et al, the inhibition of PI3k/Akt PI3k inhibitor with LY294002 (10uM) halts MK endomitosis and polyplidization, crucial steps for platelet release ( doi: 10.1242/jcs.02784). Most likely, the PI3k/Akt inhibition might reveal the same effect on platelet release from the MEG-01 cells in our in vitro system.

Finally, concerning the crosstalk between PI3K/Akt and AMPK pathway, it is well established the PI3k/AKT signaling is involved in glucose uptake and glycolysis, by also reducing AMPK activation, and in platelet release by megakaryocytes (Jaquel et al,doi:10.3390/ijms19102991; Kaushansky et al, doi: 10.1172/JCI26674; Di Buduo et al, doi: 10.3324/haematol.2016.146746).). PI3k/Akt pathway can intersect AMPK pathway at multiple points. Indeed, some regulatory sites in AMPK subunits can be phoshorylated by kinase, such as Akt, implicated in the negative regulation of AMPK. The well-established evidence about the cross-talk between PI3k/Akt and AMPK signaling and their role in megakaryopoiesis might support our findings even if we cannot establish, but only deduce, the direct cross-talk in our in vitro system.

We discussed these new references and mitigated our conclusion

Reviewer 2 Report

Unfortunately, while your study seems interesting, there are so many syntax errors and misuse of English, this reviewer found the manuscript extremely difficult to follow.  The entire manuscript need a significant rewrite.  Please employ someone that has a better comprehension of English. 

The introduction is much too general, even misleading in lines 72-75.  Megakaryocytes do not reside in the lungs as your text could implies with the statement of platelet release in the lungs.

I was happy to see a hypothesis mentioned in the introduction however, the authors did not take the opportunity to state a clear hypothesis.  Please provide a concise and clearly stated hypothesis in the introduction and list the specific aims to test that hypothesis which should then been described in the methods section.

The results section has too much text that should be incorporated in the discussion.  If you provide a clearly stated hypothesis in the introduction with subsequent specific aims, the results section should be easy to write with a a uniform flow for the manuscript.  This reviewer cannot follow the rationale of anything in the text as sections seem disjointed.  The discussion should also follow each result section appropriately.

The authors did not follow the journal guidelines by inserting figures within the manuscript as close to the mention of both Figure 1 and 2.

Figure 1 is too compressed for easy review.  Enlarge to the same size as Figure 2.  Figure legends are much too long: revise.  It may be prudent to reduce the amount of information in the figures to have them large enough for the reader to easily assess.

Page 4/18 is extremely difficult to comprehend!  Mitochondrial activity is poorly described in the methods section and should not be discussed in the results.  Line 157 is an excellent example of misuse of the English language!  The term interrogation implies that the authors had an investigative conversation with the cell line.  The reviewer understands what the authors intended (replace with: "we investigated the pathway").  Someone with a better understanding of the use of English would not have used the term.  Lines 165-166 should not be included in the results but it should be noted that regardless, the sentence is a statement without follow-up!  You speculate 2 hypotheses yet do not provide what these hypotheses might be?

Line 186 makes no sense. 

This reviewer is not criticizing your results; the manuscript is so poorly written and disjointed, I honestly have difficulty reading and trying to decipher the authors intent due to syntax errors and inappropriate use of terms/words.  I have found what I believe are some interesting findings.  The manuscript can be "saved" but requires a major re-writing as stated previously.

Author Response

While we are pleased to know that the reviewer considered our results interesting, we are sorry that he/she found many syntax English errors. As he/she suggested an extensive English revision of our manuscript, we’ve corrected manuscript thoroughly and we will also take advantage of mdpi paid editing service. The present manuscript is in fact, a revised version of the work (ijms-2113118) that we would like to be reconsidered for final publication . 

We also thank the reviewer for his/her meaningful suggestions on how to improve our scientific report. A point to point reply is here provided by the authors.

The introduction is much too general, even misleading in lines 72-75.  Megakaryocytes do not reside in the lungs as your text could implies with the statement of platelet release in the lungs.

Unfortunately, we disagree with the reviewer as a lot of recent papers clearly highlight that the lung is a site of platelet biogenesis and a reservoir for haematopoietic progenitors. Just a few references are given.

  • Pariser DN, Hilt ZT, Ture SK, Blick-Nitko SK, Looney MR, Cleary SJ, Roman-Pagan E, Saunders J 2nd, Georas SN, Veazey J, Madere F, Santos LT, Arne A, Huynh NP, Livada AC, Guerrero-Martin SM, Lyons C, Metcalf-Pate KA, McGrath KE, Palis J, Morrell CN. Lung megakaryocytes are immune modulatory cells. J Clin Invest. 2021 Jan 4;131(1):e137377. doi: 10.1172/JCI137377.
  • Yeung AK, Villacorta-Martin C, Hon S, Rock JR, Murphy GJ. Lung megakaryocytes display distinct transcriptional and phenotypic properties. Blood Adv. 2020 Dec 22;4(24):6204-6217. doi: 10.1182/bloodadvances.2020002843.
  • Lefrançais E, Ortiz-Muñoz G, Caudrillier A, Mallavia B, Liu F, Sayah DM, Thornton EE, Headley MB, David T, Coughlin SR, Krummel MF, Leavitt AD, Passegué E, Looney MR. The lung is a site of platelet biogenesis and a reservoir for haematopoietic progenitors. Nature. 2017 Apr 6;544(7648):105-109. doi: 10.1038/nature21706.
  • Zhu A, Real F, Capron C, Rosenberg AR, Silvin A, Dunsmore G, Zhu J, Cottoignies-Callamarte A, Massé JM, Moine P, Bessis S, Godement M, Geri G, Chiche JD, Valdebenito S, Belouzard S, Dubuisson J, Lorin de la Grandmaison G, Chevret S, Ginhoux F, Eugenin EA, Annane D, Bordé EC, Bomsel M. Infection of lung megakaryocytes and platelets by SARS-CoV-2 anticipate fatal COVID-19. Cell Mol Life Sci. 2022 Jun 16;79(7):365. doi: 10.1007/s00018-022-04318-x.

I was happy to see a hypothesis mentioned in the introduction however, the authors did not take the opportunity to state a clear hypothesis.  Please provide a concise and clearly stated hypothesis in the introduction and list the specific aims to test that hypothesis which should then been described in the methods section.

OK done, we’ ve better detailed our hypothesis at the end of introduction section.

The results section has too much text that should be incorporated in the discussion.  If you provide a clearly stated hypothesis in the introduction with subsequent specific aims, the results section should be easy to write with a uniform flow for the manuscript.  This reviewer cannot follow the rationale of anything in the text as sections seem disjointed.  The discussion should also follow each result section appropriately.

OK, we thank the reviewer for his/her suggestion and accordingly modified the text.

The authors did not follow the journal guidelines by inserting figures within the manuscript as close to the mention of both Figure 1 and 2.

OK, in the new revised version we embedded figures into the text, where required.

Figure 1 is too compressed for easy review.  Enlarge to the same size as Figure 2.  Figure legends are much too long: revise.  It may be prudent to reduce the amount of information in the figures to have them large enough for the reader to easily assess.

Sure, as suggested Figure 1 has been properly enlarged. Further we also tried to compress the figure legends even though the different experimental procedures require a fine description to be easily understood by the readers.

Page 4/18 is extremely difficult to comprehend!  Mitochondrial activity is poorly described in the methods section and should not be discussed in the results.  Line 157 is an excellent example of misuse of the English language!  The term interrogation implies that the authors had an investigative conversation with the cell line.  The reviewer understands what the authors intended (replace with: "we investigated the pathway").  Someone with a better understanding of the use of English would not have used the term.  Lines 165-166 should not be included in the results but it should be noted that regardless, the sentence is a statement without follow-up!  You speculate 2 hypotheses yet do not provide what these hypotheses might be?

OK, we thank the reviewer for his/her suggestion and accordingly modified the text. As he/she suggested an extensive English revisions of our manuscript, we will take advantage of mdpi paid editing service. We’ve better detailed Mitochondrial activity in M&M section and we also corrected the misuse of some words.

Line 186 makes no sense. 

OK

This reviewer is not criticizing your results; the manuscript is so poorly written and disjointed, I honestly have difficulty reading and trying to decipher the authors intent due to syntax errors and inappropriate use of terms/words.  I have found what I believe are some interesting findings.  The manuscript can be "saved" but requires a major re-writing as stated previously.

As previously said, while we are pleased to know that the reviewer considered our results interesting, we are sorry that he/she found many syntax English errors. As he/she suggested an extensive English revisions of our manuscript, we will take advantage of mdpi paid editing service.

Round 2

Reviewer 1 Report

The Authors have sufficiently replied to Reviewer's requests

Author Response

We thank again the reviewer for his/her valuable comments

Reviewer 2 Report

Thank you for defending megakaryocytes (MK) in the lungs.  I was not aware of the references.  I have taught pulmonary histology and pathology for decades and have evaluated thousands of murine and human lungs and have never seen a MK in a solitary lung.  I was a bit excited and read your sentence as if it was stating that the lungs have recently been found to be a site of thrombopoiesis rather than the term you've used: biogenesis.  I can accept this as I did read the papers you provided and in summary, they state that platelets are shed from circulating MKs that are released early from bone marrow.  

Please add a sentence to clarify the sentence as I'm sure that others will misinterpret "biogenesis".

The manuscript has been approved and syntax errors corrected

Author Response

We thank again the reviewer for his/her criticism and for the request of clarification. Indeed the term biogenesis can be really misinterpreted Now we reported in the lane 43-47 of the present manuscript the following sentence.   "Approximately half of all platelet release—a process termed thrombopoiesis occurs when megakaryocytes exit the bone marrow and enter lung capillary beds where they release platelets into the circulation via proplatelet extensions [2, 3]. Enrichment of lung MKs was described during pulmonary and cardiovascular diseases, further suggesting that they may be dynamic and responsive to inflammatory states [4, 5]...."  
